# Maintenance of Maternal Breastfeeding up to 6 Months: Predictive Models

**DOI:** 10.3390/jpm11050396

**Published:** 2021-05-11

**Authors:** Esmeralda Santacruz-Salas, Antonio Segura-Fragoso, Diana P. Pozuelo-Carrascosa, Ana Isabel Cobo-Cuenca, Juan Manuel Carmona-Torres, José Alberto Laredo-Aguilera

**Affiliations:** 1FACSALUD (Faculty of Health Sciences), Av. Real Fábrica de la seda, s/n., Talavera de la Reina, 45600 Toledo, Spain; esmeralda.santacruz@uclm.es (E.S.-S.); Antonio.Segura@uclm.es (A.S.-F.); 2Multidisciplinary Research Group in Care (IMCU), UCLM. Av. Carlos III s/n., 45071 Toledo, Spain; dianap.pozuelo@uclm.es (D.P.P.-C.); juanmanuel.carmona@uclm.es (J.M.C.-T.); josealberto.laredo@uclm.es (J.A.L.-A.); 3Department of Nursing, Physiotherapy and Occupational Therapy, University of Castilla la Mancha (UCLM), 45071 Toledo, Spain; 4Faculty of Physiotherapy and Nursing of Toledo, University of Castilla la Mancha (UCLM) Av. Carlos III s/n., 45071 Toledo, Spain; 5Maimónides Institute for Biomedical Research Córdoba (IMIBIC), 14004 Córdoba, Spain

**Keywords:** exclusive breastfeeding, breastfeeding, breastfeeding support, early breastfeeding cessation, breastfeeding difficulties, lactation, child health, motivation, weaning

## Abstract

Background: There is evidence of the benefits of exclusive breastfeeding (EBF) but maintaining EBF for the minimum recommended time of 6 months is challenging. Aims: This study aimed to determine the prevalence of breastfeeding types in a Spanish setting, explore the influencing factors, and analyze the relationships between the reasons for EBF cessation and the EBF durations achieved. Method: This longitudinal descriptive study included 236 healthy children with standard weight followed up by the public health system. A baseline survey and three telephone interviews (1, 3, and 6 months) were conducted. Results: The prevalence of EBF at 6 months was 19.49%. The mean age of the mothers was 32.3 (±5.3). The variables influencing EBF maintenance were the prior decision to practice EBF (*p* = 0.03), the belief that EBF is sufficient (*p* = 0.00), not offering water or fluid to the child (*p* = 0.04), delaying pacifier use (*p* < 0.001), a longer gestation time (*p* = 0.05), and previous experience with practicing EBF for more than 6 months (*p* = 0.00). The reason for the earliest EBF cessation (mean 52.63 ± 56.98 days) was the mother’s lack of self-efficacy (*p* = 0.05). Conclusion: Knowing the reasons for EBF cessation among mothers is important for helping mothers and preventing early weaning. A safe environment and support can prevent early weaning.

## 1. Introduction

Breastfeeding (BF) is considered the ideal diet for a newborn for both nutritional and immunological support and as a beneficial practice for the mother and child [1,2,3]. This practice not only provides the best diet but also contributes in the short, medium, and long term to the newborn’s emotional, psychological, nutritional, and developmental needs. BF maintained for more than 3 months reduces the risk of otitis media (77%), atopic dermatitis (42%), asthma (40%), and respiratory infections (75%) [4]. In addition, BF maintained for more than 6 months is associated with a 20% decrease in the risk of leukemia and a 36% decrease in the risk of sudden death [5]. The long-term benefits associated with BF, such as a lower incidence of childhood morbidity (obesity and diabetes), a two-third reduction in mortality in children aged under 5 years, and better intellectual and motor development scores in children, have been observed [6]. Among mothers who breastfeed, there is also evidence of a lower risk of certain diseases, such as obesity and breast and ovarian cancer, with an estimated reduction of almost 20,000 deaths from breast cancer [2,3,7]. The decrease in maternal and child morbidity and mortality from the practice of BF is combined with improvements in labor productivity, nonhealth costs, and environmental benefits. BF also entails significant economic savings for families and health systems. Therefore, the entire society benefits [8,9,10]. This evidence confirms the superiority of BF and supports the need to promote and preserve optimal BF practices.

Despite the health policies and programs that have been implemented, the global rates of BF are still far from the established goals. The global target for 2025 is for at least 50% of mothers to practice exclusive breastfeeding (EBF) for the first 6 months [11]. The World Health Organization (WHO) recommends BF until at least 6 months of age and its maintenance along with complementary foods until 2 years of age [12,13]. At the international level, the rates of EBF at 6 months range between 15.2% and 21% [14].

The practice of BF is a sociological phenomenon with multiple influential factors, especially during the first months. According to Rollins et al. (2016), the practice of BF can be affected by interventions in health systems, communities, and family homes; better results are observed with a combination of all three [15].

BF is an effective tool for improving overall health [13]; therefore, mothers have a high initial intention to breastfeed for the minimum recommended time. However, many mothers cease BF early [16]. In the search for information explaining this phenomenon, we spent several years studying the most influential reasons and variables. Most studies independently analyze the impact of the reasons or causes of cessation and/or sociodemographic factors that influence the onset, maintenance, or cessation of BF [17,18,19,20,21]. The results show that the most influential variables are the perception that the child is hungry, hypogalactia, use of accessories, such as a pacifier, and return to work [17,18,22,23,24]. However, we did not find many studies in our search that analyzed the relationship between the duration of EBF and/or its early cessation and the reasons for cessation mentioned by mothers [25]. The reasons are varied and depend on the time when the mothers decided to cease BF. Due to the benefits described above, it is considered necessary to further explore the factors that contribute to mothers’ success or failure in BF at each stage.

The main objective of this study was to prospectively analyze the epidemiological factors associated with the maintenance of BF during the recommended minimum period of 6 months. The secondary objectives were to (a) determine the prevalence of the different types of BF practiced in a Spanish setting and (b) determine the relationships between the reasons for EBF cessation before 6 months and the EBF durations achieved.

## 2. Materials and Methods

### 2.1. Study Setting and Sample

This prospective cohort study consecutively included women who had given birth to a healthy newborn weighing more than 2.5 kg at a gestational age between 37 and 42 weeks. The exclusion criteria were as follows: (a) abnormal health situations after delivery requiring medical examination for complications, hospital admission, or separation of the mother from the newborn; (b) multiple births; (c) positive neonatal metabolic screening results; and (d) lack of follow-up and/or health checks through the public health system. In Spain, newborn health screenings are performed through the health system for free. All newborns included in the study had to attend these visits to ensure that health professionals collected their clinical information and information related to feeding.

A minimum sample size of 230 women (46 in the EBF group and 184 in the non-EBF group) was estimated based on a pilot study conducted in our setting [10]. The sample size was calculated using the statistical program Epidat 3.1 (Galicia, v.4.1. http://dxsp.sergas.es. Accessed on 18 April 2021) with an alpha error of 5% and beta of 20%.

A flowchart of the final participant inclusion and reasons for inclusion is shown in Figure 1.

### 2.2. Study Variables

The dependent variable, i.e., EBF, was defined as an infant receiving only breast milk, except for oral rehydration solution (ORS) or drops and syrups consisting of vitamins, minerals, or medicines when necessary. Mixed feeding (MF) was defined as an infant receiving certain amounts of both breast milk and formula. Formula feeding (FF) was defined as an infant receiving exclusively formula.

The remaining variables were classified as (a) the sociodemographic and individual characteristics of the mothers and children (age, nationality, educational level, employment status, civil status, sex of the child, and maternity leave); (b) clinical variables potentially related to the practice of BF (previous number of children, gestation time, type of delivery, type of anesthesia, child birth weight, pacifier use, previous experience with BF, previous decision to practice BF, and previously decided duration of BF); and (c) variables related to the family environment or directly related to the practice of BF (believing that the child is hungry after BF, having offered bottles of water or ORS in the hospital, having offered bottles of formula, having had problems with the practice of BF, having solved the problems that occurred, reasons for introducing FF, from whom BF counseling was received, and partner’s opinion regarding BF). Among the reasons mentioned by the mothers for EBF cessation, the following were identified:(a)Cessation of EBF due to the introduction of solid foods, i.e., “by the recommendation of a health professional” or “the mother’s decision”;(b)Cessation of EBF due to the introduction of the bottle, i.e., “by recommendation of a health professional”, “influenced by the social or family environment”, “the child is not gaining weight according to the standard criteria for age”, “work reasons”, “the child is still hungry”, and” problems with breastfeeding”.

The variables “believing that the child is hungry after BF” and “the child is not gaining weight according to the standards for age” were used to identify the mother’s BF self-efficacy.

### 2.3. Measuring Procedures and Instruments

To collect the data from the sample, a personal interview was conducted with the mother during the postpartum period at the hospital using an ad hoc questionnaire. During the 6 months of follow-up, another three interviews were conducted by telephone at 1 month, 3 months, and 6 months (interval +/− 10 days) to compare the prevalence of different types of breastfeeding and collect information regarding pacifier use, return to work, and exact time of the introduction of another type of food. The remaining clinical information and/or information potentially related to the practice of BF was obtained from the electronic records of the clinical histories collected by the health workers at the follow-up visits of the newborn health program. The information collected via telephone was used to compare the prevalence of different types of breastfeeding.

### 2.4. Statistical Analysis

For the statistical analysis, SPSS Statistics v. 24.0 (IBM Corp., Armonk, NY, USA) was used. The use license belonged to the University of Castilla-La Mancha.

A descriptive analysis of the variables of interest (EBF, MF, FF, and their duration up to 6 months) was performed using counts (*n*) and proportions (%) for the qualitative variables and the mean (m) and standard deviation (SD) for the quantitative variables. The Kolmogorov–Smirnov test was conducted to determine the normality of the data, and Levene’s test was conducted to determine the homogeneity of variance. An inferential analysis was performed to determine the relationship between the variables using a chi-squared test (χ^2^) for the qualitative variables and Student’s t-test for the quantitative variables (normal distribution) or the Mann–Whitney test (nonnormal distribution). The maintenance of EBF was assessed according to the mother’s belief and supplementation of BF with water or ORS using Kaplan–Meier curves. The variables significantly associated with the maintenance of EBF for 6 months were included in the multivariate logistic regression model. An unadjusted logistic regression model was also constructed for the women who had previously had children to determine the association between previous experience with EBF and maintenance of EBF for 6 months. The odds ratios and their 95% confidence intervals were calculated. All hypothesis comparisons were two-tailed, and statistical significance was accepted at *p* < 0.05.

### 2.5. Ethical Considerations

This study was approved by the Ethics and Legislation Committee of Hospital Virgen de la Salud (CEITO. number: 74. Date: 6/06/2014). All mothers who decided to participate in the longitudinal study were informed of its content and objectives. All participating mothers signed an informed consent form.

## 3. Results

In total, 236 women and their newborns aged 0 and 6 months were included in the study. The prevalence of the different types of breastfeeding practiced by all mothers in the sample (236) observed at each of the cutoff points analyzed during the 6-month follow-up are shown in Figure 2. In total, 37 (15.7%) women practiced FF, and nine women (4.52%) practiced MF throughout the 6-month follow-up. Of the 165 women who practiced BF at some point, only 47 (19.92%) practiced BF during the full 6 months. Despite having practiced EBF, 38 (16.10%) women offered solid foods before 6 months.

The 47 (19.92%) women who practiced EBF during the 6-month follow-up formed the actual study sample. The subsequent analyses and evaluations were carried out in this group of women.

### 3.1. Maintenance of EBF

The sociodemographic and clinical characteristics of all women who practiced EBF for 6 months and their children are described in Table 1.

Some variables (“pacifier use”, “days of pregnancy”, and “duration of BF in previous children”) were analyzed independently because these variables could not be included in the multivariate model. The variable “duration of BF in previous children” did not apply to all women and only applied to those who previously had children. The other two variables were continuous and measured in days of EBF duration over 6 months (Table 2).

A multivariate analysis was carried out to analyze all significant sociodemographic and clinical variables. In the resulting predictive model, only two variables predicted the cessation of EBF before 6 months (Table 3).

The graphs shown in Figure 3 illustrate the Kaplan–Meier curves of the variables positively associated with the maintenance of EBF for up to 6 months.

### 3.2. Early Cessation of EBF

Table 4 shows the reasons mentioned by the mothers for stopping EBF before 6 months and their correlations with the duration of EBF.

Bivariate analyses were performed with each of the variables based on the different reasons mentioned by the mothers for early cessation of EBF. Variables that were found to be statistically significant were input into a multivariate analysis (Table 5).

## 4. Discussion

This study, which was carried out in a region of Spain, found that the most influential factors and determinants of the duration of EBF during the first 6 months of life of a newborn were “previous experience with EBF”, “delay in using a pacifier”, and “longer gestation time”. Many studies reported the factors influencing the maintenance of BF [17,18], but few studies explored the causes of cessation associated with the variables and reasons mentioned by the mothers [23,24,25].

### 4.1. Prevalence of the Different Types of Breastfeeding Practiced in Our Setting

The prevalence of EBF at 6 months after birth was 19.49% in our setting. This figure is well below the goals set at the international level [11,13]. According to the Breastfeeding Committee of the Spanish Association of Pediatrics, the worldwide prevalence of EBF reached 43% in 2015 [26]. In Spain, according to the latest National Health Survey (NHS) that preceded the study, in 2012, 28.5% of children received EBF until 6 months of age [27]. In the NHS conducted after our study, in 2017, it was reported that 39% of women maintained EBF for up to 6 months, which is consistent with the WHO data [28,29]. The most recent study in our setting showed a prevalence of EBF of 35.1% at hospital discharge without subsequent longitudinal data [30]. This finding implies a decrease of almost half compared to our results obtained at 1 month (66.1%). Therefore, new studies describing and comparing the current prevalence of EBF with national and international figures and describing the involved characteristics and factors are necessary.

### 4.2. Epidemiological Factors

Our study shows similarities and discrepancies with the results found in other studies regarding the epidemiological factors associated with the maintenance of EBF for 6 months.

On the one hand, we did not find statistical significance for any of the sociodemographic variables studied. Some variables, such as a younger age, lower educational level, and maternal employment status [24,31], have been found to be major obstacles to prolonged EBF in similar studies and nearby settings. In our study, similar to the cohort studied by Gipúzcoa [24], age had no statistically significant effect. However, our results are consistent with other studies regarding older mothers being able to maintain BF for longer (72.3% vs. 27.7%). This finding may be due to greater knowledge of BF and more confidence and determination when searching for helpful resources. Employment has been associated with the duration of EBF in different studies not because women have a paid job or are unemployed but because the conditions that regulate and support this practice allow women to breastfeed with ease [24,31]. However, our study agrees with other studies showing that work is one of the most important reasons cited by mothers for ceasing BF early [32].

On the other hand, in contrast to other studies, our results do not show a statistically significant association [33,34] between the type of delivery and maintenance of EBF for 6 months. Notably, compared to the women who delivered vaginally with or without instruments, almost one-third of the women who delivered by cesarean section maintained EBF for 6 months (26.4% vs. 76.6%).

Our results show that 100% of mothers who decided to practice BF before delivery managed to maintain EBF until 6 months in contrast to those who did not. Other studies also argue that the prenatal decision to breastfeed increases the chances of initiating EBF [24,35].

### 4.3. Relationship between the Reasons for Cessation and Duration of Exclusive Breastfeeding

The mothers reported several reasons for the early cessation of EBF that coincide with those found in other studies [24,32,36]. Again, our findings corroborate that women who experience problems with BF practice EBF for shorter durations. The same was observed in a study investigating the INMA birth cohort in 2015 [24]. Our findings agree with other studies that argue that maternal self-efficacy in BF is deficient and usually causes early BF cessation [37,38].

In our study, 83% of the mothers who believed that their child remained hungry after breastfeeding and 55.3% who supplemented BF with water or ORS in the hospital after delivery failed to maintain EBF for 6 months. Both subjective hypogalactia and the maintenance of these erroneous practices are decisive in the correct maintenance of BF. This finding was also reported in other studies [31,39]. In addition, these two variables or conditions were found to predict early EBF cessation in this population.

Offering bottles of formula at the beginning has also been described as a barrier to the long-term maintenance of EBF [40,41]. In our study, 57.4% of the mothers did not manage to maintain EBF for 6 months, although this result was not statistically significant.

In the final multivariate model, maternal lack of confidence with the ability to breastfeed and raise the child correctly (also called breastfeeding self-efficacy) was the most important predictor of EBF cessation. These women practiced EBF for the fewest number of days over 6 months as also reported in one of the largest studies conducted in Spain [36]. Work-related reasons along with pressure from the family and/or social environment were also predictive factors of cessation mentioned as important by mothers, which is consistent with other studies [24,32,38,42]. Similarly, in our predictive model, both reasons were decisive but not the most important. However, these reasons were associated with the longest duration of EBF and the EBF rate at 6 months.

### 4.4. Implications for Clinical Practice

Given that all women who decided before giving birth to practice BF maintained BF for 6 months, it may be relevant for health professionals to train and promote BF in childbirth preparation classes. The benefits of BF and the factors that can influence early cessation should be emphasized. In addition, it has been demonstrated that the implementation of the Baby-friendly Hospital Initiative (BFHI) is key to enhancing support to pregnant women, mothers and families and positively influencing at least the initiation of EBF [43].

Scientific evidence has shown that good training and providing help and support to mothers improves the initiation and maintenance of EBF [28,32,44,45]. Although without statistical significance, our study also showed a higher rate of EBF at 6 months (61.7%) when the mothers received advice from health professionals or BF support groups than when they did not (10.6%). Similarly, the women who did not have problems with BF at the beginning also showed higher rates of EBF at 6 months than those who did (76.6% vs. 23.4%).

### 4.5. Limitations and Strengths

The main limitations of this study are the loss to follow-up of some women in the initial sample. Because an ad hoc questionnaire was used in the study, it may not have the reliability of a validated questionnaire. However, at the time of the study, there was still no validated questionnaire for the Spanish population. The strength of this study is the fact that it was a longitudinal study with data collected from the same women over 6 months.

## 5. Conclusions

Despite the efforts, the rates of EBF at 6 months in our setting were low.

Our results show that ceasing EBF before 6 months is determined by the following two factors: having offered bottles with water or ORS to the child and maintaining the belief that the child remains hungry after BF. The long-term maintenance of EBF is related to having previously decided to practice BF, having previous experience with EBF, not carrying out practices that interfere with BF (e.g., supplementation or pacifier use), and a longer gestation time.

The main reasons for the early cessation of EBF reported by the mothers, which were significant factors, were (a) social or family environment influence, (b) work problems, (c) recommendations by health professionals, and (d) self-efficacy (believing that the child remains hungry after BF or is not gaining weight according to the standard).

It is necessary to continue evaluating whether the BF strategies carried out in our setting in recent years are effective not only for enhancing the EBF initiation rates but also for increasing the percentages of mothers who maintain EBF over time.

## Figures and Tables

**Figure 1 jpm-11-00396-f001:**
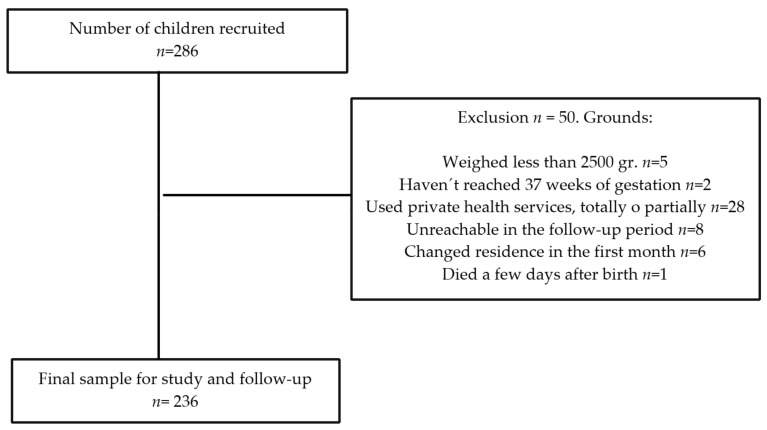
Flowchart of study participants.

**Figure 2 jpm-11-00396-f002:**
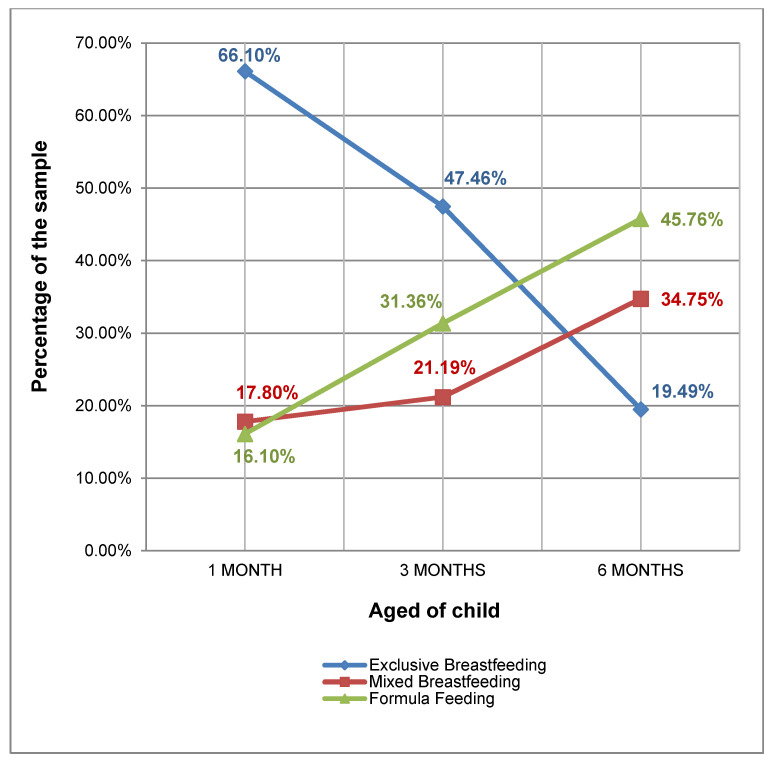
Prevalence of breastfeeding types for 6 months.

**Figure 3 jpm-11-00396-f003:**
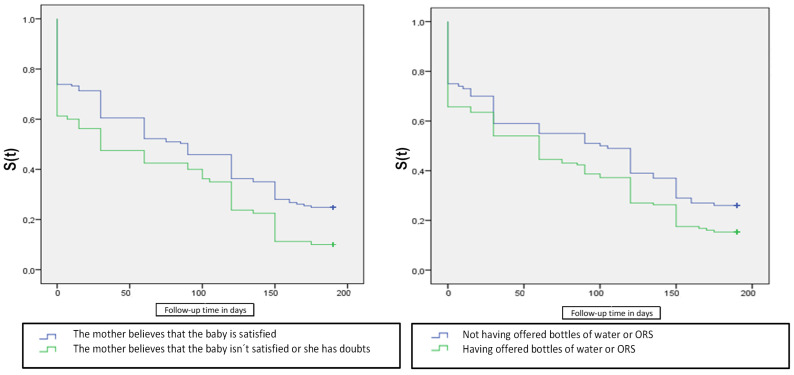
Kaplan–Meier curves of the duration of exclusive breastfeeding according to the mothers’ beliefs and supplementation with water or ORS in the breastfeeding process.

**Table 1 jpm-11-00396-t001:** Characteristics of the mother–child pairs participating in the study and their influence on the maintenance of exclusive breastfeeding for 6 months.

Variables	Not Exclusive Breastfeeding for 6 Months	Exclusive Breastfeeding for 6 Months	* *p*
	*n* (%)	*n* (%)	
Social and Individual Maternal, Infant O Mother–Child Factors			
AGE OF THE MOTHER			
Up to 30 years	57 (30.16)	13 (27.7)	0.73
>than 30 years	132 (69.8)	34 (72.3)	
NATIONALITY			
Spanish	155 (82.01)	42 (89.4)	0.23
Foreign	34 (17.99)	5 (10.6)	
EDUCATION LEVEL			
Primary school	51 (26.98)	14 (29.8)	0.68
Secondary school or higher	138 (73.02)	33 (70.2)	
LIVE WITH YOUR PARTNER			
Yes	183 (96.83)	47 (100)	0.18
No	6 (3.17)	0 (0)	
MARITAL STATUS			
Married or with partner	114 (60.32)	33 (70.2)	0.19
Other situations	75 (39.68)	14 (29.8)	
WORK SITUATION			
Employed	97 (51.32)	18 (38.3)	0.10
Unemployed	92 (48.68)	29 (61.7)	
MATERNITY LEAVE			
without maternity leave	84 (44.44)	26 (55.3)	0.17
on maternity leave	105 (55.55)	21 (44.7)	
SEX OF NEWBORN			
Boy	105 (55.55)	21 (44.7)	0.17
Girl	84 (44.44)	26 (55.3)	
CLINICAL FEATURES			
PREVIOUS NUMBER OF CHILDREN			
Neither	83 (44.15)	23 (48.9)	0.20
Only one	80 (42.33)	22 (46.8)	
Two or more	26 (13.76)	2 (4.3)	
DAYS OF GESTATION			
<280	101 (53.44)	23 (48.9)	0.55
≥280	88 (46.56)	24 (51.1)	
ANESTHESIA			
without anesthesia	28 (14.81)	5 (10.6)	0.41
with anesthesia	161 (85.2)	42 (89.4)	
BIRTH WEIGHT (kg)			
≤3.250	91 (48.1)	21 (44.7)	0.64
>3.250	98 (51.9)	26 (55.3)	
TYPE OF DELIVERY			
Vaginal with or whitout instrumental	145 (76.7)	36 (76.6)	0.97
Cesarean section	44 (23.3)	11 (23.4)	
HAVE DECIDED ON THE TYPE OF LACTATION PRIOR TO DELIVERY			
Yes, breastfeeding	173 (91.5)	47 (100)	0.03
They haven’t decided yet	17 (8.9)	0 (0.00)	
DECISION OF THE DURATION TO CONTINUE THE BREASTFEEDING			
One month	2 (1.1)	0 (0)	0.23
Two months	0	0 (0)	
Three months	2 (1.1)	0 (0)	
Between 3 and 6 month	69 (36.5)	11 (23.4)	
As long as the baby wants it	18 (9.5)	7 (14.9)	
As long as I can	86 (45.5)	29 (61.7)	
CHARACTERISTICS OF THE CLINICAL OR FAMILY ENVIRONMENT			
BELIEVE THAT THE BABY IS HUNGRY			
The baby is satisfied	118 (62.4)	39 (83)	0.00
The baby isn’t satisfied or I have doubts	71 (37.6)	8 (17)	
THEY OFFERED BOTTLES OF WATER OR ORS IN THE HOSPITAL			
Yes	73 (38.6)	26 (55.3)	0.04
Not	116 (61.4)	21 (44.7)	
THEY OFFERED FORMULA FEEDING			
Not	53 (28.1)	20 (42.6)	0.06
Yes	136 (71.9)	27 (57.4)	
INITIAL PRACTICE PROBLEMS WITH BREASTFEEDING			
Not	155 (82.1)	36 (76.6)	0.38
Yes	34 (17.9)	11 (23.4)	
THEY SOLVED THE BREASTFEEDING PROBLEMS			
Not	81 (42.9)	23 (48.9)	0.47
Yes	108 (57.1)	24 (51.1)	
BREASTFEEDING COUNSELLING			
Not recivied or not necesary	75 (39.7)	13 (27.7)	0.18
Search itself	25 (13.2)	5 (10.6)	
Health profesional or support group	89 (47.01)	29 (61.7)	
COUPLE OPINION REGARDING BREASTFEEDING			
Not in favor	3 (1.6)	0 (0)	0.08
In favor	171 (90.5)	47 (100)	
Indifferent	15 (7.93)	0 (0)	

ORS = oral rehydration solutions; * *p*. = statistical significance.

**Table 2 jpm-11-00396-t002:** Relationship between the other variables and the maintenance of exclusive breastfeeding for 6 months (women who practiced exclusive breastfeeding for some time during the 6 months).

Relationship beteween Use of Pacifier and Time Achieved of EBF				
Variables	*n* (%)	Time of EBF (Days)	SD	χ^2^
USE OF THE PACIFIER				
Not use pacifier	68 (28.8)	118.9	69.14	<0.001
Start using in the first 29 days	67 (28.4)	57.19	67.42	
Start using from the first 30 days	101 (42.8)	73.37	74.28	
RELATIONSHIP BETEWEEN AVERAGE NUMBER OF DAYS OF PREGNANCY AND TIME ACHIEVED OF EBF				
VARIABLES	*n* (%)	Time of pregnancy (days)	SD	t
EBF 3 MONTHS				
NOT	124 (52.5)	275.23	8.6	0.001
YES	112 (47.45)	277.93	7.6	
EBF 6 MONTHS				
NOT	190 (88.5)	275.99	8.47	0.05
YES	46 (19.49)	278.64	6.79	
**RELATIONSHIP BETEWEEN PRIOR EXPERIENCE IN EBF AND MAINTENANCE OF EBF 6 MONTHS**				
VARIABLES	*n* (%)	Maintenance of EBF 6 months (OR)	CI 95%	*p**
TIME OF EBF TO OTHERS PREVIOUS CHILDRENS				
No time	4 (7.14)	Ref.		
Until 6 months	7 (13.73)	2.205	(0.59–8.22)	0.239
Over 6 months	14 (40)	8.902	(2.60–30.4)	0.00

EBF = exclusive breastfeeding; SD = standard deviation; χ^2^ = Chi-square test; t = Student t-test; OR = Odds ratio; CI = Confidence Interval; *p** = statistical significance.

**Table 3 jpm-11-00396-t003:** Conditions influencing the cessation of exclusive breastfeeding at 6 months.

Influential Variables	OR	IC 95%	* *p*
BELIEVE THE BABY IS NOT SATISFIED OR HAVING DOUBTS	2.96	(0.149–768)	0.01
HAVING OFFERED BOTTLES OF WATER OR ORS	0.52	(1.000–3.707)	0.05

OR = Odds Ratio, CI = Confidence interval, * *p* = statistical significance.

**Table 4 jpm-11-00396-t004:** Reasons for cessation and duration of exclusive breastfeeding (women who practiced exclusive breastfeeding).

Reasons Aluded by the Mothers		*n* (%)	Average Days of EBF	SD
Reasons for introducing artificial milk bottles	Health recommendation	7 (2.9)	115.7	41.3
	Influence of social or family environment	11 (4.6)	107.8	51.5
	The baby does not weight gain according to standards	4 (1.6)	97.5	58.1
	Labour problems	24 (10.1)	96.0	40.2
	The baby gets hungry	21 (8.9)	76.7	51.9
	Problems with breastfeeding	27 (11.4)	60.8	49.8
Reasons for introducing solid foods	Health recommendation	84 (35.6)	100.4	55.1
	She decides for herself	7 (2.9)	105.7	60.2

EBF = exclusive breastfeeding, SD = standard deviation.

**Table 5 jpm-11-00396-t005:** Reasons cited by mothers for ceasing exclusive breastfeeding. Multivariate analysis (women who practiced exclusive breastfeeding some time for 6 months).

Reasons for Introducing Artificial Milk Bottles	*n* (%)	Average Days of EBF	SD	* *p*
Influence of social or family environment	162 (68.6)	88.9	81.3	0.05
Labour problems	26 (11)	88.65	46.53	
Health recommendation	11 (4.7)	73.64	66.56	
Autoeficacy = The baby gets hungry + not sufficient weight gain according to standards	38 (16.1)	52.63	56.98	

EBF = Exclusive Breastfeeding, SD = standard deviation, * *p* = statistical significance.

## Data Availability

The data that support the findings of this study are available from the corresponding author, upon reasonable request.

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
