# Peer review of "Maintenance of Maternal Breastfeeding up to 6 Months: Predictive Models"

_jpm, 2021, doi:10.3390/jpm11050396_

Round 1
Reviewer 1 Report
This manuscript addresses the extremely important topic of “exclusive breastfeeding” and its great benefits for individuals and society with valuable impacts on health and economic savings. Also it notifies that the global target for 2025 of at least 50% of mothers to practice exclusive breastfeeding in the first 6 months of the newborn’s life will not be accomplished, as the international rates are very far from the goals set. In this way, the manuscript gathers new insights and knowledge about epidemiological and influential factors associated with the maintenance of breastfeeding plus the obstacles reported by mothers to fulfill the recommended minimum period of 6 months of exclusive breastfeeding.
The authors should be prepared to incorporate some major revisions. Also, english language must be improved specially in ‘Discussion’ section.
Introduction provides a good explanation of the topic and is very clear, as well as the sections ‘Materials and Methods’ and ‘Results’. However the flowchart of the study participants (figure 1) and the description of the sample characteristics that appears in ‘Results’ section should be changed to the ‘Materials and Methods’.
Nevertheless, I found the ‘Discussion’ section to lack clarity and a clean language. The language used is not always clear and would benefit from further editing and revision. My main recommendation is to rephrase almost all the ‘Discussion’ section (beginning in line 241), and the last two paragraphs of ‘Conclusion’ section, which I think will enhance the clarity of the paper.
Finally, references 10 and 26 are the same work:
- Santacruz-Salas E, Aranda-Reneo, I., Hidalgo-Vega, Á., Blanco-Rodriguez, J. M., & Segura-Fragoso, A. The Economic Influence of Breastfeeding on the Health Cost of Newborns. Journal of Human Lactation [Internet]. 2018. Available from: https://doi.org/10.1177/0890334418812026.
- Santacruz-Salas E, Aranda-Reneo I, Hidalgo-Vega Á, Blanco-Rodriguez JM, Segura-Fragoso A. The economic influence of breastfeeding on the health cost of newborns. Journal of Human Lactation. 419 2019;35(2):340-8.
Author Response
Dear reviewer:
Enclosed you will find a new revised version of our manuscript, “MAINTENANCE OF MATERNAL BREASTFEEDING UP TO 6 MONTHS: PREDICTIVE MODELS” (manuscript number: jpm-1178032). We would like to thank you for giving us the opportunity to revise and improve our manuscript; we also thank the reviewers for their thoughtful and constructive comments. We have considered all the suggestions and have incorporated them into the revised manuscript, and as a result we believe our manuscript is stronger. An itemized point-by-point response to the comments is presented below.
Thank you for your attention in this matter.

Reviewer 2 Report
Line 58: in the first 6 months of the newborn’s life PLEASE DELETE newborn’s IS REALLY IN THE FIRST 6 MONTHS (WHEN IS NO NEWBORN ANYMORE].
Line 59: This period is the minimum time that the WHO recommends maintaining EB The recommendation for EBF is 6 months (WHO), not minimum time, as it also recommends starting complementary foods at 6 months of age while continuing breastfeeding up to 2 years of age or beyond
Line 60: WHO also recommends maintaining BF supplemented with other foods until 2 years of age I suggest to use the term "with complementary foods" According to different publications BMS could be considered supplements (and use to replace breast milk)
Lines 81-82 maintenance of BF during the recommended minimum period of 6 months. I suggest separating EBF and BF.... It is true that many women stop BF at all early; but is also true that many start other foods or artificial milk early in the infancy, for the reasons indicated by the authors; in this case they do not complete the recommended time of EBF but are still breastfeeding their babies
Lines 181-183 There were 37 (15.7%) women who practiced EB and nine women (4.52%) who practiced MF during the 6-month follow-up. Of the 165 women who practiced BF at some point, only 47 (19.92%) practiced it during the full 6 months I suggest reviewing this as at the beginning is indicated that 37 women were exclusively breastfeeding, with 9 practicing mixed feeding and the final part (group considered the exclusive breastfeeding mothers) the number is 47.
Line 185 The 47 (19.92%) women who practiced BF during the 6-month follow-up. Please use the term exclusive breastfeeding (EB) otherwise it seems like they were practicing any breastfeeding.
Lines 213-214 The graphs in Figure 3 show the survival curves for the variables positively associated with the maintenance of EB for up to 6 months. I find the graph confusing; first why do we talk about survival? Maybe is more relation with the practice? And it seems that duration was lower when mothers did not offer bottles of water or ORS than those that did offer that.
GENERAL: I find the study very interesting, with a few questions that are expressed above. It would be good to indicate that the definitions of exclusive breastfeeding and other practices are used looking at behavior the last 24 hours (as the question is presented to mother in surveys) and in mothers that are followed the definition is stricter and hence the numbers and percentages are usually lower; that seems to be the situation in this study where mothers were followed-up. Could be useful to clarify that point.
Another point worth to stress is the facility environment, it has been demonstrated that the implementation of BFHI (Baby-friendly Hospital Initiative) is key to enhance support to pregnant women, mothers and families and influences at least initiation and exclusive breastfeeding (although there is also data on relation of duration, seems that we need more information to reach conclusions) It seems that this factor (baby-friendliness) was not considered by the authors, maybe they can acknowledge that?
Author Response
Dear Reviewer:
Enclosed you will find a new revised version of our manuscript, “MAINTENANCE OF MATERNAL BREASTFEEDING UP TO 6 MONTHS: PREDICTIVE MODELS” (manuscript number: jpm-1178032). We would like to thank you for giving us the opportunity to revise and improve our manuscript; we also thank the reviewers for their thoughtful and constructive comments. We have considered all the suggestions and have incorporated them into the revised manuscript, and as a result we believe our manuscript is stronger. An itemized point-by-point response to the comments is presented below.
Thank you for your attention in this matter.

Reviewer 3 Report
The manuscript is interesting and covers a highly relevant subject however I find it being outside the scope for J Pers Med. This is for the editor to decide.
Other points I would like to comment:
*Addresses are preferably described using English terms so the reader could understand the author affiliations.
*The Abstract is clearly written with clear Aims followed up in Results. This clear organisation is not found in the rest of the manuscript where Objectives are described using other wording. I believe the manuscript could benefit from being more clear connecting Aim with objectives and results.
*The Abstract Background:”…types in our setting..” I believe “in a Spanish setting” or likely would be more clear to use in the short abstract.
*The authors have done a good job including many relevant references, however many (12,13,19,22,17,30…) are in Spanish and thus not available to the English-speaking scientific community. PDFs are difficult to run through google translate…
*Row 100 - Please explain what Epidat 3.1 is
*Fig 1 – The lower left box seem to be misplaced
*Fig 1 number of excluded patients: 5+2+28+8+6 = 49 not 50 !! Please clarify
*Fig 2 – The graphical design could be improved ( Number of value figures, Time of breastfeeding = Age of child ?, describe in figure legend the meaning of EB/MB/FF) IS MB = MF?
*When discussing fig 2 in text the numbers in the text and the numbers in fig2 does not match and the legends MB (?) makes this very hard to understand for the reader. However when reading discussion 4.1 it is more clear.
*Table 1 “live with partner” 183+47 does not add upp to 236. Please explain, also the next is not 236. I Have not test calculated all but see that also the lactation question has similar issue.
*table 1 Birth weight in milligram? You must mean kg?
*Table 3 – Typ0 “dubts”, “ods” should be “Odds ratio”
*Fig 3 – “mum” is not an appropriate word in this context. Use Mother. “Hot having offered” ? Do you mean “not”? The axis legends are so small in font size that they cannot be read – is it up to 200,0 days on X-axis? Why is it given with a decimal = 4 number of values? I would prefer that months were used as in Fig 2.
*Table 4 – Some figures are given with five (!) value figures – eg 115.71 days. One decimal is sufficient in my opinion.
*Table 5 – “No weight gain” – I interpret as there is no weight gain at all. I guess you mean “not sufficient weight gain”
*I guess it was not possible to reach the mothers exactly on the day 1/3/6 months due to holidays or other engagements. What interval was accepted interval +/- 7 days or other?
*Row 346/347, 349-51 all contain Spanish and must be translated
Author Response

(The authors gave the same response as above.)

Round 2
Reviewer 3 Report
The authors have adjusted the manuscript accordingly.